# Short Tandem Repeat-Enriched Architectural RNAs in Nuclear Bodies: Functions and Associated Diseases

**DOI:** 10.3390/ncrna6010006

**Published:** 2020-02-20

**Authors:** Kensuke Ninomiya, Tetsuro Hirose

**Affiliations:** Institute for Genetic Medicine, Hokkaido University, Sapporo 060-0815, Japan; k-ninomiya@igm.hokudai.ac.jp

**Keywords:** architectural RNA, nuclear body, short tandem repeat-enriched RNA, repeat-expansion disease, toxic RNA

## Abstract

Nuclear bodies are membraneless, phase-separated compartments that concentrate specific proteins and RNAs in the nucleus. They are believed to serve as sites for the modification, sequestration, and storage of specific factors, and to act as organizational hubs of chromatin structure to control gene expression and cellular function. Architectural (arc) RNA, a class of long noncoding RNA (lncRNA), plays essential roles in the formation of nuclear bodies. Herein, we focus on specific arcRNAs containing short tandem repeat-enriched sequences and introduce their biological functions and recently elucidated underlying molecular mechanism. In various neurodegenerative diseases, abnormal nuclear and cytoplasmic bodies are built on disease-causing RNAs or toxic RNAs with aberrantly expanded short tandem repeat-enriched sequences. We discuss the possible analogous functions of natural arcRNAs and toxic RNAs with short tandem repeat-enriched sequences. Finally, we describe the technical utility of short tandem repeat-enriched arcRNAs as a model for exploring the structures and functions of nuclear bodies, as well as the pathogenic mechanisms of neurodegenerative diseases.

## 1. Introduction

Nuclear bodies are membraneless compartments in the nucleus, ~0.2 to 3 μm in diameter, which usually exhibit phase-separated features and liquid-like properties. In mammalian cells, there is a wide variety of nuclear bodies, including the nucleolus, Cajal bodies, histone locus bodies (HLBs), nuclear speckles, nuclear stress bodies (nSBs), paraspeckles, the perinucleolar compartment (PNC), polycomb bodies, and promyelocytic leukemia bodies. Nuclear bodies are defined by enrichment of specific marker proteins and RNA molecules, and some are conditionally formed upon experiencing certain stresses [1,2,3,4]. A subset of nuclear bodies are built on specific scaffolding long noncoding RNAs (lncRNAs) which include paraspeckles, nSBs, and PNCs formed on nuclear paraspeckle assembly transcript 1 (NEAT1), highly-repetitive satellite III (HSATIII), and pyrimidine-rich noncoding transcript (PNCTR), respectively [5,6,7]. These lncRNAs with membraneless nuclear body scaffolding functions have been termed architectural RNAs (arcRNAs), since they represent a distinct subcategory among thousands of lncRNAs [8,9]. The existence of multiple types of novel RNA-dependent nuclear bodies has been confirmed in mammalian cells [10,11]. LncRNAs with similar architectural functions have also been identified in Drosophila and fission yeast [12,13,14], suggesting that arcRNAs are widely utilized as structural scaffolds to form membraneless organelles in all eukaryotic species. In the case of the arcRNAs defined above, one specific arcRNA acts as a scaffold for building its cognate membraneless organelle. However, in other membraneless organelles, such as cytoplasmic stress granules, it has recently been reported that multiple RNA species are sequestrated to form a scaffold; therefore, in this case, complex RNA–RNA and RNA–protein interactions are involved in the building of these structures [15,16].

The biological roles of nuclear bodies remain poorly understood, but it has been proposed to be classified them into three functional categories [4,17]. First, they act as a reaction crucible in which specific molecules, such as enzymes and their substrates, accumulate, and thereby catalysis is facilitated. For example, nuclear speckles are widely considered a site for phosphorylation of serine/arginine-rich splicing factors (SRSFs) [18]. HLBs concentrate FLICE-associated huge protein (FLASH), histone cleavage complex (HCC), and U7 small nuclear RNP to promote efficient histone mRNA processing in a cell cycle-dependent manner [19]. Secondly, they act as a molecular sponge in which specific proteins and RNAs are stored and protected from degradation, or sequestrated from nucleoplasm, to negatively regulate their nucleoplasmic functions. Paraspeckles are believed to act as molecular sponges; upregulation of NEAT1 leads to an increase in the number and size of paraspeckles, resulting in sequestration of splicing factor proline- and glutamine-rich (SFPQ) into enlarged paraspeckles separate from the nucleoplasm. Consequently, SFPQ-mediated transcription of target genes such as *ADARB2* and *IL8* is downregulated [20]. Thirdly, they act as an organizational hub that anchors multiple chromatin loci, presumably to integrate gene expression at these loci. Super enhancer (SE) is a cluster of enhancers in which high densities of transcription coactivators accumulate to form a phase-separated condensate [21]. Thus, SE is believed to enhance the efficiency of transcription. Paraspeckles are also reported to anchor multiple transcriptionally activated promoters [22,23], raising an intriguing possibility that they can act as an organizational hub for chromatin loci, as well as a molecular sponge as described above.

## 2. Short Tandem Repeat-Enriched ArcRNAs

To implement these functions, especially the reaction crucible and molecular sponge functions, a subset of arcRNAs possess repeat sequences to accumulate multiple copies of specific proteins and/or RNAs. For example, the cytoplasmic noncoding RNA activated by DNA damage (NORAD) lncRNA sequestrates PUMILIO via at least 17 binding sites with an eight nucleotide (nt) unit (UGUAUAUA) to repress its function, which is essential for maintenance of genomic stability and mitochondrial function [24,25,26]. Xist lncRNA, which plays a critical role in forming the Xist cloud during X chromosome inactivation, contains several functional repeats (e.g., an A repeat unit with 25 nt and a C repeat unit with ~120 nt) [27,28]. Two 5′ SnoRNA capped and 3′ polyadenylated (SPA) lncRNAs sequestrate TDP43, RNA binding Fox-1 homolog 2 (RBFOX2), and heterogeneous nuclear ribonucleoprotein M (HNRNPM) through the repetitive UG-rich sequence to regulate splicing [29,30]. The nuclear lncRNA functional intergenic repeating RNA element (Firre) interacts with nuclear matrix protein heterogeneous nuclear ribonucleoprotein U (HNRNPU) via 156 nt repeats (16 in mouse and eight in human) to control nuclear architecture [31]. The Drosophila arcRNA heat-shock RNA-omega (Hsr omega) contains tandem repeats of 280 nt in a stretch of ~10 kb that contribute to the recruitment of various RNA-binding proteins to omega speckles (thermal stress-induced nuclear bodies) [12,14]. The yeast arcRNA meiRNA forms nuclear foci called Mei2 dots at its own genomic locus and contains at least 25 UNAAAC repeats that sequester the RNA-binding protein (RBP) Mmi1 to inhibit its function and stimulate the progression of meiosis [13].

Since most canonical RNA-binding domains of RBPs recognize five to six nt motifs for specific binding [32,33], arcRNAs possessing multiple tandem repeats of short sequence stretches, representing a small subcategory of arcRNAs, are more advantageous for efficiently sequestering a large number of specific RBPs with high affinity to an RNA molecule with multiple repeat sequences (Table 1). YAP et al. (2018) screened RNA sequencing (RNA-seq) reads consisting of short-tandem repeats that had been excluded from conventional transcriptome analyses because they were non-uniquely mapped. By collating with the RNA-binding motif database, functional short-tandem repeat-enriched RNAs were predicted, eventually yielding five RNAs that are physiologically expressed in human cells as candidate functional short-tandem repeat-enriched RNAs [6]. Among them, pyrimidine-rich noncoding transcript (PNCTR) RNA contains multivalent CUCUCU-hexamers that bind to pyrimidine tract-binding protein 1 (PTBP1). PNCTR acts as a functional arcRNA by serving as an essential scaffold for PNC and sequestering PTBP1 within PNCs separate from the nucleoplasm to inhibit its splicing regulatory function and promote cell survival.

## 3. Nuclear Stress Bodies Formed by HSATIII ArcRNAs

HSATIII is an additional example of a short tandem repeat-enriched arcRNA that forms nuclear stress bodies (nSBs), consisting mainly of highly-repetitive (GGAAU)n sequences [41]. The primate-specific pericentromeric satellite III regions are transcriptionally silent under normal conditions, but are transcribed to synthesize HSATIII arcRNAs under thermal stress conditions [5,42,43]. HSATIII arcRNAs remain stable in nuclei, but form membraneless nuclear stress bodies (nSBs) upon recruitment of specific RNA-binding proteins such as Scaffold attachment factor B (SAFB), specific sets of SRSFs, transcription factors HSF1 and CREBBP, and bromodomain protein BRD4 [5,36,37,44]. In our recent study, comprehensive proteomics analysis of HSATIII ribonucleoprotein complexes, which are most likely nSB complexes, using the chromatin isolation by RNA purification-mass spectrometry (ChIRP-MS) technique identified 141 previously unreported proteins, many of which were nuclear RNA-binding proteins involved in pre-mRNA splicing and processing [34]. In addition, cell division cycle (CDC)-like kinase 1 (CLK1), a nuclear protein kinase that phosphorylates SRSFs, is recruited to nSBs specifically during the recovery phase after thermal stress (Figure 1). During thermal stress, SRSF1 and SRSF9 are dephosphorylated and recruited to nSBs. After thermal stress, they are rapidly rephosphorylated by CLK1 kinase, and consequently, which enables rapid changes in pre-mRNA splicing [34]. Thus, nSBs are dynamic nuclear bodies, the composition of which changes in response to temperature shifts that serve as a reaction crucible for efficient SRSF phosphorylation, which is required for temperature-dependent rapid control of pre-mRNA splicing (Figure 1).

Among nSB-localized SRSFs and SR-related proteins, SRSF1 and SRSF9 are particularly highly enriched in nSBs, and share the common GGARG binding motif (R is A or G) [32], which is similar to the repeats in HSATIII arcRNAs (Figure 1). This strongly suggests that SRSF1 and SRSF9 bind directly to the enriched GGAAU repeat sequence in HSATIII, resulting in a high local concentration in nSBs.

HSATIII arcRNAs also serve as a scaffold for a minor subclass of nSBs (named nSB-M) including HNRNPM, which is distinct from the major population of HSATIII-dependent bodies characterized by SAFB enrichment (named nSB-S) [35]. This suggests that nSBs are heterologous nuclear bodies with distinct structures and functions. Intriguingly, nSB-S and nSB-M are formed on common HSATIII arcRNAs with different sets of RNA-binding proteins. The molecular mechanism of the formation of the two distinct nSBs with HSATIII arcRNAs remains to be investigated.

## 4. Comparison of nSBs and Spinocerebellar Ataxia Type 31 (SCA31) Foci Formed by Analogous Short Tandem Repeat-Enriched ArcRNAs

Spinocerebellar ataxia type 31 (SCA31) is a neurological disease caused by insertion of a 2.5 to 3.8 kb repeat that includes the long TGGAA stretch present in a common intronic region of BEAN1 (brain expressed associated with NEDD4-1) and thymidine kinase 2 (TK2) genes [39,40,45]. BEAN1 transcripts containing expanded UGGAA repeats are mainly retained in nuclei where they form RNA foci in Purkinje cells of the cerebellum of SCA31 patients. These RNA foci sequester a number of RNA-binding proteins including TDP43 (Table 1). This is possible because the UGGAA tandem repeat contains multiple copies of GAAUG, which is identical to the high-affinity binding motif of TDP43 [32]. A subpopulation of SCA31 repeat RNA is believed to be transported to the cytoplasm and translated into a toxic pentapeptide repeat (PPR), which is a potential cause of the disorder [39]. Moreover, phenotypic analysis using Drosophila models of SCA31 expressing long UGGAA repeat RNAs revealed toxic effects that are suppressed by the RNA chaperone-like function of TDP43, FUS, and HNRNPA2/B1. Additionally, nontoxic short UGGAA repeat RNA suppresses the toxic effects of the ALS-causing TDP43 mutation. This raises an intriguing possibility that SCA31 repeat RNAs and ALS-causing factors such as TDP43, FUS, and HNRNPA2B1 could interact to neutralize the respective toxic effects, suggesting that the disruption induced by aberrant expansion of the SCA31 repeat and the TDP43 mutation could be responsible for the neurodegenerative disease phenotypes [39]. Interestingly, the SCA31 UGGAA repeat is essentially equivalent to the GGAAU repeat in HSATIII arcRNAs. It is possible that SCA31 foci control pre-mRNA splicing, which is analogous to the recently discovered function of HSATIII in nSBs described above. Indeed, the HSATIII-interacting proteins identified by ChIRP-MS partially overlap with those of the UGGAA repeat RNA identified by in vitro binding experiments (Table 1) [34,35,39,40], suggesting that at least the overlapping proteins, such as TDP43 and FUS, interact directly or indirectly with GGAAU repeats rather than with non-repeat regions. It should be noted that HSATIII complexes are nSBs that are specifically formed under thermal stress conditions, but SCA31 RNA foci are formed under normal temperature conditions. Thus, RNAs possessing analogous repeat sequences potentially form distinct RNP complexes with partially overlapping protein components, and play distinct or partially overlapping roles depending on the context, environment, and tissue type (Figure 2).

## 5. Future Perspectives

In addition to SCA31 RNA described above, recent studies have demonstrated that several kinds of short-tandem repeat-enriched arcRNAs are also synthesized from specific gene loci containing pathogenic expansion of three to six nt repeats, which are believed to cause various neurological and neuromuscular disorders such as myotonic dystrophy 1 and 2, fragile X tremor ataxia syndrome (FXTAS), SCA8, SCA10, and SCA36 [46,47,48]. Hence, these repeat RNAs are termed toxic RNAs. Since the disease-associated repeat regions are, in most cases, located within introns or 5′- or 3′-UTRs of host genes, they should function as part of pre- or spliced mRNAs (or possibly excised introns) transcribed from host genes. The proposed pathogenic mechanism involves toxic RNAs sequestering specific sets of RNA-binding proteins to form abnormal aggregates, which likely impairs cellular functions and generates toxic repeat polypeptides by AUG-dependent or repeat-associated non-AUG (RAN) translation [49,50]. However, in the case of most disease associated repeat RNAs, a comprehensive overview of their binding proteins and the precise pathogenic mechanisms has not been performed.

Functional analysis of short tandem repeat-enriched arcRNAs has provided important insight into the structures and functions of membraneless nuclear bodies formed on specific arcRNAs. Additionally, this class of arcRNAs can serve as models for structural and functional analysis of arcRNAs that are difficult to investigate directly due to technical reasons. Firstly, short tandem repeat-enriched arcRNAs are highly specific for particular nuclear bodies, hence they are the most reliable markers of nuclear bodies, whereas most associated proteins are typically located in the nucleoplasm outside nuclear bodies. In most cases, these arcRNAs are efficiently knocked down by nucleofection of a single antisense oligonucleotide (ASO) that hybridizes with multiple sites of the tandem repeat stretch, which enables the specific disruption of nuclear bodies and the ability to investigate their functions. Hybridization-based techniques for RNP precipitation, such as ChIRP, capture hybridization analysis of RNA targets (CHART), and RNA antisense purification (RAP) [51], have been developed and utilized to purify the components of RNA-based complexes. These methods are powerful tools for the comprehensive identification of factors associated with specific RNAs in vivo, but it is usually difficult to design appropriate antisense oligonucleotides (ASOs) that can act as probes for efficient capture of specific RNP complexes, presumably due to inefficient association of ASOs that target RNAs, which are largely masked by proteins. In the case of repeat RNAs, ASOs can more efficiently capture entire RNP complexes, presumably through multivalent hybridization with some unmasked regions of target RNAs. Furthermore, efficient and stable purification of substantial amounts of RNPs allows us to perform biochemical analysis of time course- and signal-dependent changes within RNP complexes, such as exploring changes in phosphorylation states of specific components of RNPs [34]. For the same reasons described above, this class of arcRNAs are relatively easily detected by fluorescent in situ hybridization (FISH). Furthermore, since databases that assort binding sequence motifs of multiple RBPs have been constructed [32,33], proteins that bind the specific repeat region can be predicted using bioinformatics. A subset of novel arcRNAs, whose binding proteins remain identifiable, also possess repeat sequences [10].

Therefore, this approach is applicable for unveiling the structures and functions of novel arcRNA-dependent cellular bodies and disease-related RNA foci. Currently, small compounds that specifically bind to disease-related repeat RNAs have been explored as candidate therapeutic drugs [52,53,54]. Further research should be directed to evaluate such candidate drugs in disease model cells and investigate their modes of action. Intensive investigation of short tandem repeat-enriched arcRNAs provides important insights into the pathogenic mechanisms of RNA repeat expansion-related neurological and neuromuscular diseases, and reveals some basic concepts pertinent to therapeutic development.

## Figures and Tables

**Figure 1 ncrna-06-00006-f001:**
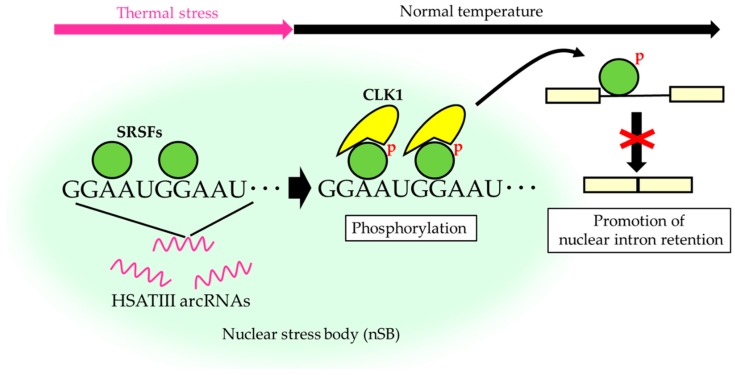
The reaction crucible model of nuclear stress bodies (nSBs) for serine/arginine-rich splicing factors (SRSF) phosphorylation. SRSFs are globally dephosphorylated upon thermal stress exposure. Specific SRSFs, including SRSF1, SRSF7, and SRSF9, are concentrated within nSBs through association with highly-repetitive satellite III (HSATIII) arcRNAs. Cell division cycle (CDC)-like kinase 1 (CLK1) is specifically recruited to nSBs during recovery after thermal stress cessation to rapidly rephosphorylate nSB-localized SRSFs, which represses splicing of hundreds of specific introns.

**Figure 2 ncrna-06-00006-f002:**
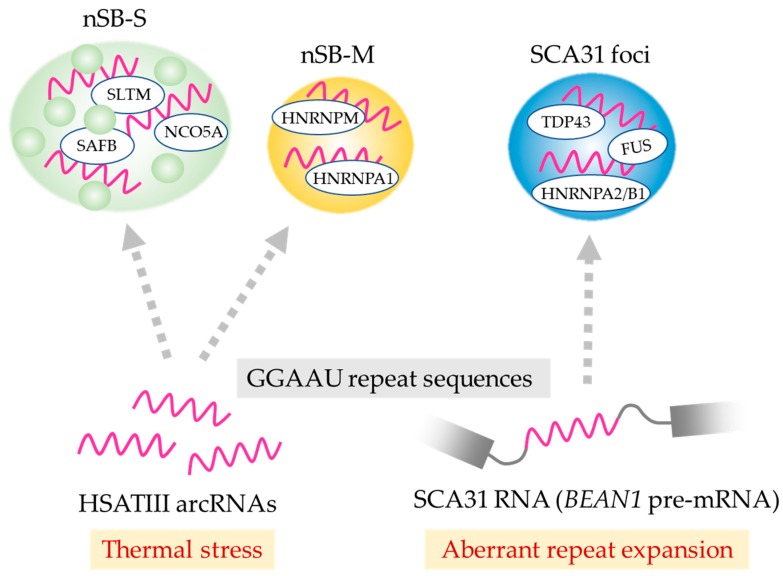
Distinct nuclear bodies assemble on a common repeat sequence of arcRNAs. nSB-S and nSB-M are mutually exclusive subnuclear bodies formed on HSATIII arcRNAs upon thermal stress exposure. SCA31 foci are formed on the expanded UGGAA repeat region of BEAN1 transcripts in the nuclei of Purkinje cells in the brains of SCA31 patients. The identified resident proteins are indicated in each nuclear body. GGAAU repeat-enriched regions are colored magenta.

**Table 1 ncrna-06-00006-t001:** List of short tandem repeat-enriched RNAs.

ArcRNA	Enriched Repeat	Nuclear Body	Binding Proteins	Functions and Roles	References
PNCTR	CUCUCU	PNC	PTBP1	Control of pre-mRNA splicing through sequestration of PTBP1	[6]
HSATIII	GGAAU	nSB-S and/or nSB-M	More than 100 proteins including SRSF1, SRSF9, CLK1, TDP43, FUS, SFPQ, CREBBP, BRD4, HSF1	Induced upon thermal stress. Control of pre-mRNA splicing through re-phosphorylation of SRSFs during recovery after stress removal	[5,34,35,36,37,38]
		nSB-S	SAFB, SLTM, NCOA5		
		nSB-M	HNRNPM, HNRNPA1, HNRNPH1		
SCA31 repeat	UGGAA	SCA31 foci	SRSF1, SRSF9, TDP43, HNRNPA2/B1, FUS, SFPQ	Expanded in SCA31 patients. Unusual production of toxic pentapeptides	[39,40]
SPA	UG-rich sequence	PWS region body	TDP43, RBFOX, HNRNPM	Deleted in PWS patients. Sequestration of RBPs to regulate splicing	[29,30]
meiRNA	UNAAAC	Mei2 dot	Mmi1	Promoting the progression of meiosis by repressing Mmi1	[13]

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
