# Peer review of "Short Tandem Repeat-Enriched Architectural RNAs in Nuclear Bodies: Functions and Associated Diseases"

_ncrna, 2020, doi:10.3390/ncrna6010006_

Round 1

Reviewer 1 Report

In this review Ninomiya and Hirose discuss Architectural (arc) RNAs , a
class of long non-coding RNAs (lncRNA) which plays essential roles in the formation of nuclear bodies. They focus on specific arcRNAs, consisting of short tandem repeat-enriched sequences, for which biological functions have been recently discovered. Moreover, they discuss the possible analogous functions of natural arcRNAs and toxic RNAs that can be a tool to investigating the structures and functions of nuclear bodies, as well as the pathogenic mechanisms of neurodegenerative diseases. 

A great, interesting and timely review that provides innovative perspectives to comprehension of the function of  non-coding transcripts with a low complexity repetitive nature, particularly for their impact in modulating the  nuclear function 

Minor point

The authors may consider to include references within Table 1 for each of the listed ArcRNA.  

Author Response

The authors may consider to include references within Table 1 for each of the listed ArcRNA. 

-> We added Reference into Table 1.

Reviewer 2 Report

This is a nice and timely review on the recent progress of short repeat-enriched RNAs related to nuclear bodies and diseases. I have the following points for the authors to consider.

The architectural function is one of the multiple roles of many short tandem repeat-enriched RNAs presented in the review. As this review focuses on the short tandem repeat-enriched RNAs in nuclear bodies, especially nSBs and SCA31 foci, the current title is somehow too broad to precisely reflect the main messages included in the text. I therefore suggest the authors to mention these key points in the title, for example, they may consider to add “Nuclear Bodies” in the title, and change the current title as “Short Tandem Repeat-Enriched RNAs in Nuclear Bodies: Functions and Associated Diseases”. If the authors wanted to keep the current title as is, it would be nice to include some general features of how to classify and annotate short tandem repeat-enriched RNAs. It is nice that the authors have discussed the diversity of this kind of RNAs across species, including NORAD, Firre, Hsr omega, meiRNA, PNCTR. But several well-studied RNAs, such as NORAD, Firre, Hsr omega are absent in Table 1. Further, a few other important repeat-enriched RNA should be included. For examples, Xist contains several functional repeats (RepA) and can form an Xist cloud during XCI (PMID:28068310; PMID:18974356); the PWS (Prader-Willi syndrome) region sno-lncRNAs and SPAs contain short tandem repeats for RbFox2 and TDP43 binding, respectively and form a nuclear accumulation in normal cells to impact splicing (PMID:27871485; PMID:22959273). In the following section, the authors have reviewed nSBs and SCA31 foci and the related short tandem repeats RNAs. A couple of references should be included. Page 3, line 90. The reference that describes the repeat element (GGAAU)n in HASTIII RNA is needed. For example, the recent annotation of HASTIII sequence using biotinylated oligonucleotides (PMID: 15788562). Page 3, line 92. Transcription of HSATIII is a general stress response in human cells (PMID: 18039709), it would be better to note that the thermal stress is a classical method to induce HSATIII transcription. Page 6, line 173. This sentence should be read as “… tandem repeat-enriched arcRNAs has provided important insight…”

Author Response

The architectural function is one of the multiple roles of many short tandem repeat-enriched RNAs presented in the review. As this review focuses on the short tandem repeat-enriched RNAs in nuclear bodies, especially nSBs and SCA31 foci, the current title is somehow too broad to precisely reflect the main messages included in the text. I therefore suggest the authors to mention these key points in the title, for example, they may consider to add “Nuclear Bodies” in the title, and change the current title as “Short Tandem Repeat-Enriched RNAs in Nuclear Bodies: Functions and Associated Diseases”. If the authors wanted to keep the current title as is, it would be nice to include some general features of how to classify and annotate short tandem repeat-enriched RNAs. It is nice that the authors have discussed the diversity of this kind of RNAs across species, including NORAD, Firre, Hsr omega, meiRNA, PNCTR. But several well-studied RNAs, such as NORAD, Firre, Hsr omega are absent in Table 1. Further, a few other important repeat-enriched RNA should be included. For examples, Xist contains several functional repeats (RepA) and can form an Xist cloud during XCI (PMID:28068310; PMID:18974356); the PWS (Prader-Willi syndrome) region sno-lncRNAs and SPAs contain short tandem repeats for RbFox2 and TDP43 binding, respectively and form a nuclear accumulation in normal cells to impact splicing (PMID:27871485; PMID:22959273).

-> According to the reviewer’s suggestion, the title was modified by adding “in nuclear bodies” (line 3). Xist and SPAs were included as additional repeat-enriched ncRNAs (line 71-76). The property of SPA was included into Table 1.

In the following section, the authors have reviewed nSBs and SCA31 foci and the related short tandem repeats RNAs. A couple of references should be included. Page 3, line 90. The reference that describes the repeat element (GGAAU)n in HASTIII RNA is needed. For example, the recent annotation of HASTIII sequence using biotinylated oligonucleotides (PMID: 15788562). Page 3, line 92. Transcription of HSATIII is a general stress response in human cells (PMID: 18039709), it would be better to note that the thermal stress is a classical method to induce HSATIII transcription. Page 6, line 173. This sentence should be read as “… tandem repeat-enriched arcRNAs has provided important insight…”

-> The references (Ref #43, 44, 45) describing the HSATIII property were included (line 100~103).

-> The sentence was modified according the reviewer’s suggestion (line 183).

Reviewer 3 Report

The manuscript is devoted to the role of short tandem repeat-enriched architectural RNAs (arcRNAs) in nuclear domains formation and associated diseases. The authors present a list of currently known short tandem repeat-enriched architectural RNAs and their biological functions with special emphasize on human HSATIII arcRNA. The major strength of discussion is comparison of nSBs and SCA31 foci formed by analogous short tandem repeat-enriched arcRNAs. These findings are very important and interesting. 

There are some minor points to be considered such as these:

The terms describing the shape of nuclear bodies mentioned in the manuscript ("spheroidal" and "granular") are not very suitable, since many nuclear domains do not have granular or spherical appearance.  Section "Future perspectives": please describe in more details the suggested approach mentioned in line 198. Some sentences are very long and difficult to understand, for example: "SRSF1 and SRSF9 are recruited to nSBs during thermal stress, where they are rapidly rephosphorylated by CLK1 kinase that is recruited immediately after stress removal, and this consequently enables implementation of rapid changes in pre-mRNA splicing when stress is alleviated following a shift to normal temperature [28]." line 100: "(CDC)-like kinase 1 (CLK1), a nuclear protein kinase that forms SRSFs" forms should be changed to phosphorylates. line 50: " up-regulation of NEAT1 leads increases the number" should be either leads to or increases.

Author Response

The terms describing the shape of nuclear bodies mentioned in the manuscript ("spheroidal" and "granular") are not very suitable, since many nuclear domains do not have granular or spherical appearance. 

-> These words were deleted from the text (lanes 9 and 26).

Section "Future perspectives": please describe in more details the suggested approach mentioned in line 198.

-> The sentence describing a possible approach for therapeutics was added in line 209-212.

Some sentences are very long and difficult to understand, for example: "SRSF1 and SRSF9 are recruited to nSBs during thermal stress, where they are rapidly rephosphorylated by CLK1 kinase that is recruited immediately after stress removal, and this consequently enables implementation of rapid changes in pre-mRNA splicing when stress is alleviated following a shift to normal temperature [28]."

-> This long sentence was divided into two sentences (line 111~114).

line 100: "(CDC)-like kinase 1 (CLK1), a nuclear protein kinase that forms SRSFs" forms should be changed to phosphorylates. line 50: " up-regulation of NEAT1 leads increases the number" should be either leads to or increases.

-> They were corrected.

Reviewer 4 Report

This review is novel and timely in that it focuses on recently characterized components common to a number of lncRNAs that are capable of nucleating phase-separated structures referred to as nuclear microbodies. It is generally very well written and contains numerous new ideas and observations. That said, there are some potentially misleading or confusing generalizations and omitted considerations that should be dealt with prior to publication.

First, it should be pointed out in the introduction that most arcRNAs have various types of repetitive elements or features, and how the short tandem repeats (STRs) discussed here are unique with respect to other repeated elements. The authors should also consider/acknowledge that STRs may not always be necessary or sufficient for arcRNA activity. Most (all?) arcRNAs also contain longer repeats. Repeats in general may be a functionally unifying feature of RNAs with lipid-lipid phase separation (LLPS) structure-forming activity. It’s also possible or likely that many non-tandem repeats/elements can be brought within proximity of one another by RNA or DNA folding.

It should also be considered that the current definition of arcRNAs may be limiting, simplistic or deceptive in that, as implied above, many other RNAs with different types or numbers of repeats may also have arcRNA properties, but may not be fully sufficient on their own or when in low concentrations. This may be the case for arcRNAs involved in Cajal body, nuclear detention centre, polycomb body, super enhancer, stress granule etc LLPS structures. Such caveats should be considered and discussed prior to delving into STRs specifically. Indeed, some of these features may contribute to the heterogeneity of the nuclear microbodies discussed in the review that are nucleated by a common species of STR-containing arcRNA. Other relevant and common features of arcRNAs not discussed are their unusual sizes, and regulation by differential splicing (Krause, TIGs 2018). In light of the many new types of LLPS structures being discovered (many of which are not listed in this review (see for ex Alberti et al, Cell, 2019), focusing on a limited number of nuclear microbody structures that happen to use arcRNAs with STRs, without acknowledging others, may be misleading. Perhaps this is part of what distinguishes nuclear microbodies from other, more irregular LLPS structures.

Although the authors have covered this before, it would also be helpful/appropriate to discuss what’s currently known about interactions between arcRNAs and PLD/LCD domain-containing TFs. Many of these proteins are listed in the review, but how they interact and with which sequences is not. For example, is it known whether they interact with STRs. If so, are the STRs both necessary and sufficient? 

The section on involvements of arcRNAs in disease is good, but the authors could expand more on possible disease treatment possibilities and difficulties.

The reference list is not very extensive and missing some highly relevant citations.

The following are small typos etc:

line 13: change consisting to containing. Consisting implies exclusivity.

line 50: “leads increases”. 

Multiple usage of abbreviations without initial or appropriate clarification: ie RBP, PNC, nSB (nuclear stress body - line 93), SRSFs, etc

line 129: expressed associated

line 137/138: add references

line 173: provide

Author Response

First, it should be pointed out in the introduction that most arcRNAs have various types of repetitive elements or features, and how the short tandem repeats (STRs) discussed here are unique with respect to other repeated elements. The authors should also consider/acknowledge that STRs may not always be necessary or sufficient for arcRNA activity. Most (all?) arcRNAs also contain longer repeats. Repeats in general may be a functionally unifying feature of RNAs with lipid-lipid phase separation (LLPS) structure-forming activity. It’s also possible or likely that many non-tandem repeats/elements can be brought within proximity of one another by RNA or DNA folding.

-> Although we had described the longer repeats in arcRNAs, we added more information of the repeat motifs in XIST and SPA ncRNAs (line 71~76) (related to Reviewer 2). Also it clarified that strRNA is just a small subclass of arcRNAs (line 85~86).

It should also be considered that the current definition of arcRNAs may be limiting, simplistic or deceptive in that, as implied above, many other RNAs with different types or numbers of repeats may also have arcRNA properties, but may not be fully sufficient on their own or when in low concentrations. This may be the case for arcRNAs involved in Cajal body, nuclear detention centre, polycomb body, super enhancer, stress granule etc LLPS structures. Such caveats should be considered and discussed prior to delving into STRs specifically. Indeed, some of these features may contribute to the heterogeneity of the nuclear microbodies discussed in the review that are nucleated by a common species of STR-containing arcRNA. Other relevant and common features of arcRNAs not discussed are their unusual sizes, and regulation by differential splicing (Krause, TIGs 2018). In light of the many new types of LLPS structures being discovered (many of which are not listed in this review (see for ex Alberti et al, Cell, 2019), focusing on a limited number of nuclear microbody structures that happen to use arcRNAs with STRs, without acknowledging others, may be misleading. Perhaps this is part of what distinguishes nuclear microbodies from other, more irregular LLPS structures.

-> We clearly described the defined arcRNA solely plays the function to build a specific membraneless organelle, which is comparable with the other organelle such as cytoplasmic stress granule containing multiple RNA species (line 41~45).

Although the authors have covered this before, it would also be helpful/appropriate to discuss what’s currently known about interactions between arcRNAs and PLD/LCD domain-containing TFs. Many of these proteins are listed in the review, but how they interact and with which sequences is not. For example, is it known whether they interact with STRs. If so, are the STRs both necessary and sufficient?

-> The detail molecular mechanism underlying the interactions between arcRNAs and TFs (we are not sure what TFs mean, it may be “trans factor”) remains poorly investigated. In most cases, what is known are that the certain RBP binds specific arcRNAs. The binding specificity of these RBPs have not been investigated. Hence, we added a sentence speculating the interaction of two overlapped RBPs (TDP43 and FUS) with GGAAU repeat (line 156-158).

The section on involvements of arcRNAs in disease is good, but the authors could expand more on possible disease treatment possibilities and difficulties.

-> A sentence regarding small chemical compound that targets strRNAs is added to mention the possibility of strRNAs as therapeutic targets (line 209-212).

The reference list is not very extensive and missing some highly relevant citations.

-> Several references related to the newly added sentences were included in Reference (related to Reviewer 2).

The following are small typos etc:

line 13: change consisting to containing. Consisting implies exclusivity.

line 50: “leads increases”.

Multiple usage of abbreviations without initial or appropriate clarification: ie RBP, PNC, nSB (nuclear stress body - line 93), SRSFs, etc

line 129: expressed associated

line 137/138: add references

line 173: provide

-> All the minor points above were corrected in the text.

Round 2

Reviewer 1 Report

The authors have completely addressed my concerns 

Reviewer 2 Report

The revised MS has addressed my concerns and I support the publication of this MS.

Reviewer 3 Report

The manuscript can be accepted for publication in its present form.

Reviewer 4 Report

I think the authors have responded well to the reviewer suggestions and that the review is essentially fine as is.

Note that 'TF' referred to transcription factors.